# Assessment of COVID-19 vaccination-related medical waste management practices in Bangladesh

**Md Rayhanul Islam Rayhan[1], Jannatul Mawya Liza[1], Md. Mostafizur Rahman[1,2]***

**1** Department of Environmental Sciences, Jahangirnagar University, Dhaka, Bangladesh, **2** Laboratory of Environmental Health and Ecotoxicology, Department of Environmental Sciences, Jahangirnagar University, Dhaka, Bangladesh

* rahmanmm@juniv.edu

**Data Availability Statement:** All relevant data are within the paper.

**Funding:** The author(s) received no specific funding for this work.

## Abstract

The COVID-19 pandemic forces people to be vaccinated as early as possible. The COVID-19 vaccination program certainly raised the medical waste volume all over the world, including in Bangladesh. Numerous recent reports showed a fragile medical waste management system in Bangladesh; during the pandemic, the situation became worse. In addition, the nation-wide ongoing COVID-19 vaccination processes have been posing an extra burden to the existing biomedical waste management in the country. Failing to proper management of this waste might be a threat to human and environmental health. Therefore, the study investigated the current COVID-19 vaccine waste management practices in Bangladesh and made a comparison to the proposed standard operating procedures of international organizations and vaccine waste management practices of two other countries (USA and India). The study was carried out through a mixed methodological approach such as qualitative and quantitative, including a questionnaire survey in 15 Upazila of 4 Districts (Dhaka, Narayanganj, Manikganj, and Gazipur) of Bangladesh. The article focused on a nation-wide legitimate COVID-19 vaccination waste estimation, strength, weakness, opportunity, and threat (SWOT) analysis and drivers, pressure, state, impact, and response (DPSIR) framework analysis to identify the present state of medical waste management in the study area. The study found an excellent segregation system (100%) but very poor waste handling (35.5%) along with very poor syringes and sharps disposal method (open burning without buried 46.6%) and poor vials disposal method (without disinfection/open dump 52%) of vaccine waste. It is estimated that about 58 and 257.85 tonnes of syringes (with needles and packaging) and vaccine vials (Sinopharm 2 doses) waste have been generated since the mass-vaccination program started. Upon SWOT analysis, good separation techniques, poor waste management (ex-situ), enough space for management, and environmental and human health concerns were mostly identified as a strength, weakness, opportunity, and threat, respectively. Finally, a DPSIR framework was prepared for vaccine waste generation and its consequences in the studied area. This study will be useful to prepare a suitable vaccination waste management system in Bangladesh.

**Competing interests:** The authors have declared that no competing interests exist.

## 1. Introduction

COVID-19 continues to ravage Bangladesh and is already infecting 15.7 M people, causing over 28,028 deaths between the initial outbreak between 8th March 2020 and December 2021 [1]. With the growing number of infections caused by COVID-19, the amount of medical waste has also increased. Manila, Kuala Lumpur, Hanoi, Bangkok, and some cities in the UK generate more waste than before the pandemic, and the estimated waste generation is now around 154–280 tonnes per day than before the pandemic [2]. An estimated 1.63–1.99 kg of medical waste in Dhaka was generated per bed per day before COVID-19. Recent study found that daily, 206 to 250 tonnes of medical waste are generated due to COVID-19 alone in the capital city of Bangladesh [3, 4]. Infection due to COVID-19 has created huge waste pressure. It has been reported that COVID-19 medical waste from the infected patient was 658.08 tonnes in March 2020, which increased dramatically in April 2021 and turned into 16164.74 tonnes [5]. Due to the potential health hazards of the COVID-19 virus, the demand for vaccination to develop immunity is of great importance now. So far, 4 different manufacturers/ brands of COVID-19 vaccines have been used in Bangladesh. Those are Astrazeneca, Pfizer, Sinopharm, and Moderna. Among them, Pfizer and Moderna are mRNA COVID-19 vaccines, AstraZeneca is viral vector vaccines, and Sinopharm is inactivated whole-virus vaccines [6]. The country has targeted vaccinating its 80% population (approx. 117,856,000) above 18 years of having a National Identity (NID) card. Already 62.04% (approximately 85.7 M) of targeted people completed their first doses, and 49.6% (approximately 42.06 M) have completed both $1^{st}$ and $2^{nd}$ doses till 12-December -2021 [7]. As the medical waste management system in Bangladesh has already been damaged by COVID-19 disease, it is crystal clear that the mass vaccination program is imposing another huge burden on the current medical waste management system in developing countries like Bangladesh [5].

Vaccinations can take place at hospitals, both on-site and off-site (tents, pop-ups, annex buildings, and other temporary venues), as well as retail pharmacies, clinics, and long-term care institutions. Waste is generated throughout the life cycle of vaccination and the application stage. The main types of waste generated from a vaccination program are Vials, Syringes, Sharps (Needles), Plastic packets (which contain sharps and syringes), PPE, and packaging materials (Plastic, cardboard, paper) [8]. The Vaccine itself contributes both in terms of open and closed vial waste. It has been observed that between 0.3 and 30% of waste accounts for COVID-19 vaccination [8, 9]. For the production of the vials, the glass melted with 4% moisture at a 1500˚C temperature, and it requires intense energy. It consumes around 0.98 TWh of energy per year. The furnace, which is used to melt the glass responsible for CO2 gas emissions (around 650 kt/y). Production of syringes also requires a high range of energy (around 223 GW per $1.56 \times 10^{10}$ of doses) [9]. Not only their production requires much energy but also their management when turning into wastage, requires intense energy. The incineration method can manage syringes because of its high purification and sterilization capacity. The incineration process needs a high temperature for performing its function, and as a result, it requires high energy. Non-incineration techniques can also be used to treat syringes [10]. If Vaccine related wastage is not managed in a proper way, this can create a looming waste crisis and severe impacts on both humans and the environment.

Although the doses of Vaccines could be handled as non-hazardous, their management is associated with environmental issues [8, 11]. While lockdown due to COVID-19 lowered the air pollution and greenhouse gas emission, Vaccine related wastage increases the energy consumption through their production and management [8]. They also increase the amount of solid and liquid waste. Through high energy consumption and waste generation, vaccination programs have an impact on air, water, and soil pollution [8].

According to WHO, all discarded Vaccine related wastage must be safely collected, treated, transported, and disposed of. Used COVID-19 vaccine vials and associated materials are considered infectious materials, having potential risks to human health. The waste management of COVID-19 vaccination must be under the supervision of well-trained staff responsible for making a plan prior to launching the vaccination activity [12]. Stericycle, a US-based B2B company, suggests increasing the armory of reusable sharps containers to reduce waste load and not displace the container from other patients' areas [13]. Moreover, CDC declared that Manufacturers of each Vaccine should have provided proper guidelines on how to dispose of the waste generated from the vaccination programs [14]. WHO, Stericycle, CDC, and other organizations provide a standard management system for managing vaccine waste. By following these management systems, vials, syringes, and packaging materials can be treated, and the detrimental effect of these materials on the environment and human health can be minimized. Thus, the current study reviewed these management systems and their implications in the management practice of COVID-19 vaccine waste so that the policymakers of Bangladesh can prepare an effective vaccine waste management plan.

Bangladesh has a total of 8 divisions. Under 8 divisions, there are 64 Districts. Districts are divided into an administrative subunit which is known as Upazila. There are a total of 495 Upazila. Each Upazila contains a health complex unit from which vaccination programme is being performed.

Before the vaccination program started, training was given to the Medical Technologists-Expanded Programme on Immunization (MT-EPI) of 495 Upazila on how the vaccination program will be continued under the patronage of the Government of Bangladesh. A part of this training was related to waste management entitled "Waste Removal and Things to Do after Session." This part suggests burning all the wastes except the vials in a 3×3×5 sized pit and then burying the ash. The vials are recommended to crash in a sack, submerge in a chlorine mixture (Bleaching powder), and then be buried in the disposal pit [15].

Previously a lot of research work [2, 16–20]. was performed on biomedical waste management issues in the District level medical special emphasis on the capital city. But there is no evidence of how the health complex at the Upazila level manages its induced waste. This study investigated 15 Upazilas in four Districts. The number of Covid-19 cases is high among these Districts, and 3 of them stood among the top 10 Districts by confirmed cases. Besides this, the vaccine waste of central Dhaka is managed by PRISM Bangladesh Foundation, whereas there is no proper waste management system in these 15 Upazilas. T. Chowdhury et al. 2021 has reported an estimation of vaccine waste, but they have the limitation of taking only a single weight of vials. There is a difference in the weight of vials between the different brands of Vaccines. We have estimated all the possible vial wastes by weighing each brand of vaccine vials. So, the current study will explore the current management practices of medical waste at the Upazila level in Bangladesh with a special emphasis on COVID-19 vaccination-related wastes.

## 2. Materials and methods

### 2.1 Study area

The study was carried out in 15 Upazilas of 4 Districts (Dhaka, Gazipur, Manikganj, and Narayanganj) of Bangladesh. The data was collected from the Upazila Health Care Center. A list of the study points and their geographic locations is presented as follows (Fig 1). The study area map was prepared by using QGIS software [21]. The study area map was produced using QGIS software [21] and the base map was obtained from open accessed internet source- Central Intelligence Agency [22].

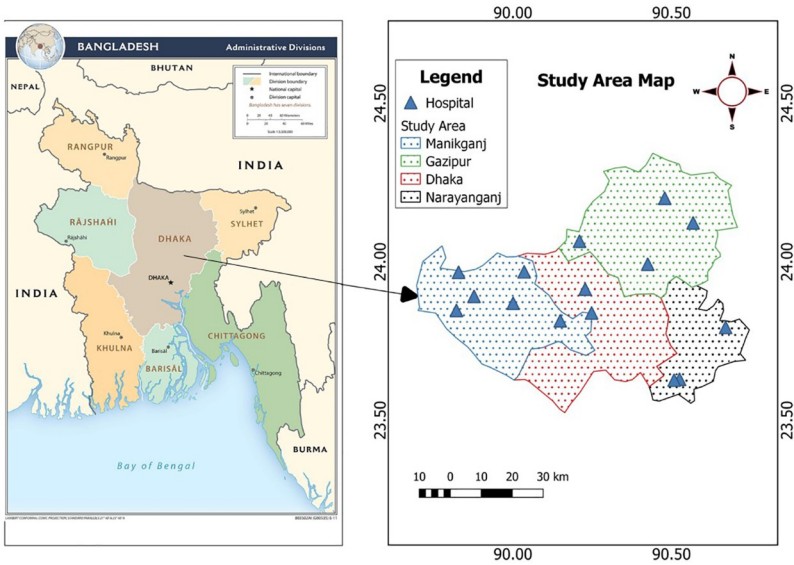

**Fig 1. Map of the study area.**

## 2.2 Secondary data analysis

The current policy of vaccine waste management, suggested by the international organizations (WHO, CDC), International Agency (Stericycle, BWS), Ministry of Health and Family Welfare (MOHFW) of Bangladesh governments, and another country (India) was reviewed from different journals and websites. The existing management and management plan of Bangladesh was reviewed from the website of the Department of Environment (DoE), MOHFW, COVID-19 vaccination guideline, which was provided to the MT(EPI) before the vaccination program started [15, 23].

## 2.3 Questionnaires

A questionnaire survey through the interview was conducted using close-ended questionnaires, which were prepared following WHO, CDC, and other guidelines. The interview was directed to the Medical Technologist of EPI (MT-EPI) or/and EPI Superintend who were appointed to manage Vaccine-related waste of the respective Upazila Health Complex. The questionnaires had 4 parts.

1. General Information (Person and Job Information):
   Name, Age, Sex, Job location, and their role in the management of COVID-19 vaccine waste.

2. Rudimentary arrangement and policy for management of hospital waste:

   - How much did they know about policy, regulation, and plan for Medical Waste Management?

   - Did they segregate hospital waste?

   - How did they store the hospital waste after segregation?

   - What was the ultimate fate of the waste?

3. Current Practices in the management of COVID-19 vaccination-related waste:

- Did they maintain separate guidelines for the management of COVID-19 vaccine waste?

- What was the ranking of the maintenance of these separate guidelines in that hospital?

- Did they receive any training for the maintenance of the vaccine waste?

- Did they segregate vials, syringes, sharps, and packaging materials after finishing the vaccination activity?

- Did they count the vials? If yes, what was the number of vials per day?

- What were the packaging materials for collecting Vials, PPE, Cotton, Wraps?

- Did they collect syringes and sharps in a safety box?

- After collecting vials and syringes, in what place did they store them?

- How much time did they store this waste?

- What was the transportation procedure to collect waste from storage to the treatment site?

4. Ongoing Treatment policy for management of COVID-19 Vaccine waste:

- What were the safety measurements before handling the vaccine waste?

- What were the treatment procedures to treat this waste?

- Did they receive sharps in a proper safety box?

- What did they do with the Syringe and sharps?

- Did they sterilize vials before disposal?

- Before final disposal, did they crush the vials?

- What was the ultimate fate of these vials, Syringe, and sharps?

## 2.4 Calculation of positive management practice

The positive management practice of the study area was assessed based on WHO-provided Standard Operating Procedure (SOP) [12]. The factors that denote positive practice are as shown in the following Table 1.

The percentage of positive management practice was calculated by using the following formula:

$$M_{\%p} = \left[ \sum F_{\%m} \times \left( \sum F_{\%s} \right) \right] \tag{i}$$

Where,
$M_{\%p}$ = Percentage of the Positive Management Practice
$F_{\%m}$ = Percentage of the Main Factors
$F_{\%s}$ = Percentage of the Sub-Factors

## 2.5 Estimation of vaccination waste generation

The total waste estimation was split into two parts. In both parts, the weight of the vaccination materials was measured. The weight of the empty vial is the same for each brand. Thus we just collect a single vial for each brand as a sample. Also, the weight of the Syringe and associated packaging material is the same all over the country.

**Table 1. Factors considered for positive management practice.**

| Management Practices | Factors indicate the positive practice |
|---|---|
| Vaccine Waste Segregation | • Well-managed segregation plan of vaccination waste.<br>• Maintaining separate boxes for vials and sharps |
| Handling of the waste | • Wearing Face Mask, PPE, Heavy Duty Gloves, Boots during the treatment process. |
| Safety Measures in Management activities | • Wearing Face Mask, PPE, Hand Gloves during management activities in health complexes. |
| Storage Facility | -Maintaining a secured and locked room:<br>• *Accesses only Designated authority.<br>• *Protection from Sunlight, Rain, Water, Food, Rodents, Plague. |
| Disinfection status | • Disinfection of the waste using Chlorine-compounds. |
| Policy Management | • Maintaining separate guidelines to manage vaccine waste |

*Subfactors under the main factors.

In the first part, the weight of the plastic Syringe and associated packaging materials was measured by an electric weight machine (Model: JJ224BC, Sl No. 142417043025). The weight of these materials is then multiplied by the total number of vaccine doses. As a result, total waste generated from Syringe and associated packaging materials were estimated.

$$S_w = W_s \times \{N_1 + (N_2 \times 2)\} \tag{ii}$$

Where,

$S_w$ = Total weight of waste generated from Syringe and associated packaging materials

$W_s$ = Weight of a single syringe, needle, needle cap, and packaging materials

$N_1$ = Total number of completions of 1st doses

$N_2$ = Total number of completions of 2nd doses

The average amount of Syringe and sharp waste generation per month was also calculated.

$$S_{wm} = \frac{T_{sw}}{T_{nm}} \tag{iii}$$

Where,

$S_{wm}$ = Total weight of Syringe and sharp wastes generated per month

$T_{sw}$ = Total amount of Syringe and sharp wastes generated by vaccination activity.

$T_{nm}$ = Total number of months of vaccination activity.

The predicted Syringe and sharp wastes against the targeted vaccination people also estimated using the following formula:

$$S_{pw} = W_s \times N_t \times 2 \tag{iv}$$

Where,

$S_{pw}$ = Predicted weight of the wastes generated from Syringe and Sharps (Including the packaging)

$W_s$ = Weight of a single syringe, needle, needle cap, and packaging materials

$N_t$ = Targeted population to be vaccinated (Multiplied by 2 as each Vaccine contain 2 doses)

In the second part, the weight of the vaccine vials was measured by the same weight scale. The weight of a single vial was multiplied by the total number of doses and divided by the

doses contained by the vials (as some vials contain more than one dose).

$$V_w = \frac{N_1 + N_2}{n} \times W_v \qquad \text{(v)}$$

Where,
$V_w$ = Total weight of the waste generated from vaccine vials
$W_v$ = Weight of a unit vaccine vials
n = Number of doses in a single vial
$N_1$ = Total number of completion of only 1st doses
$N_2$ = Total number of completion of only 2nd doses

However, prediction of the amount of waste generated from only vials against the targeted population to be vaccinated could not be possible due to the unavailability of exact information.

## 2.6 Strength-Weakness-Opportunity-Threat (SWOT) analysis of current waste management practice

Strength, Weakness, Opportunities, and Threat is one of the high skills for strategic analysis. Both environmental relationships and the development of appropriate paths for countries, companies, organizations, and other entities can be achieved through SWOT analysis [24, 25].

*Strength*: An internal intensifier of resources, attributes, and competence.

*Weakness*: This is also an internal factor but works as a constraint for resources, attributes, and competence that is important for success.

*Opportunities*: An external intensifier that can be pursued for the acquirement of benefit.

*Threats*: This external factor works as an inhibitor that potentially reduces accomplishments [26].

SWOT analysis can be used for 2 types of companies one is health care, government organizations, and non-profitable companies, and another one is for profitable companies [27]

The SWOT analysis of the studied area was done on the basis of the questionnaire survey and visual observation.

## 2.7 Drivers-Pressures-States-Impacts-Responses (DPSIR) framework for the medical waste management in the healthcare facilities

DPSIR is one of the most exclusive frameworks to demonstrate the link between social and environmental risk factors to human health [28]. This framework was first adopted by Rapport and Friend and later was adopted by the Organization for Economic Cooperation and Development (OECD) and the European Environmental Agency (EEA) [29]. According to the DPSIR framework, there is a chain of causal links starting with 'driving forces' (economic sectors, human activities) through 'pressures' (emissions, waste) to 'states' (Physical, chemical, and biological) and 'impacts' on ecosystems, human health, and functions, eventually leading to political 'responses' (prioritization, target setting, indicators) [30].

The close-ended questionnaire survey and site observation are the basis of creating a DPSIR framework in the studied area.

## 3. Results and discussions

### 3.1 Reviewing existing guidelines on COVID-19 vaccine waste management: Global and national perspectives

As part of the objectives, the study reviewed some guidelines regarding COVID-19 vaccination waste management introduced by many International Organizations, waste management agencies of a reference country (United States of America), and the Government of Bangladesh. The reviewed guideline depicts in the following table (Table 2)

### 3.2 Findings on existing management practices

A schematic flow diagram of the current management practices is shown in the following figure (Fig 2), which is also described later.

A good segregation technique of Vaccine related waste was observed in all of the studied health complexes. After using the syringes containing sharps, 93.33% of the users put them in a paper-made safety box provided by the Government. We found users put the syringes with sharps in an open paper-made container, which is not a safety box in Manikganj Sadar Hospital. They confessed to the scarcity of the safety box provided by the Government. 60% of the health complexes stored the used vials in a cartoon (which is made from paper and also used for the packaging of the syringe & Vaccine related materials). Only 13.33% of the health complexes kept the vials in an enclosed plastic container (Plastic Bin), recommended by WHO to store used vaccine vials. 26.66% of the health complexes used single plastic sacks to store the vials.

After the vaccination programme of each day, staff led by the medical technologist of EPI (MT-EPI) count the total vaccine doses by counting the number of vials. When the counting was over, the vials were then stored. 80% of the health complexes stored in a secured and locked room, whereas 13.3% of the health complexes stored the vials in an open place, mainly under the stairs of the building or a corner on a floor of the building. According to WHO, the used vaccine vials and syringes must be stored in a secured and locked room which must be protected from sunlight, rodent, plagues, other foods, and other staff [12]. Considering all of these factors, the study found 76% of positive practices in the studied health complex, which scaled as a good practice (Table 3).

In Table 3, management practices were defined by the result of the questionnaire survey that was made on the basis of WHO guidelines. For each management practice (Column 1), single or multiple questions were asked by the related authority of vaccine waste management (MT-EPI). The mathematical average of the percentages of each positive response (factors) was denoted as good practice (Column 2). The specific definition of each practice (Table 3) are outlined as follows:

Vaccine waste segregation = vaccine-related waste segregation, vials, and syringe separation.

Handling of the waste = Wearing face mask, Personal Protective Equipment (PPE), Gloves, Heavy-duty gloves, Boots in the waste disposal unit.

Safety measures in Management = Using masks, PPE, and Surgical Gloves in the hospital.

Storage Facility = Secured and locked room- protection from sun, water, other food sources, rodents, and other staff.

Disinfection status = Using chlorine-related compound

Policy management = Separate guidelines for vaccine waste disposal.

**Table 2. Guidelines regarded to the treatment of the discarded COVID-19 vaccine waste.**

| Guidelines and References | | Vaccine related Wastages | | |
|---|---|---|---|---|
| | | Syringe and Sharps | Vials | Other Materials |
| International Organizations | WHO [12] | • Syringe and sharp must be collected in a dedicated safety box.<br>• After collection, they should be discarded in a dedicated sharps pit. This should be done without removing the waste from the safety box. No need to sterilize this waste before disposal.<br>• After that, the sharps and Syringe should be incinerated in a proper incinerator. An incinerator with a high temperature and a double gas chamber to remove toxic gas is preferable.<br>• The ash generated from incineration should be disposed of in a dedicated ash pit which was previously identified by the health authority. | • Vials should be collected in a leak-proof bag (Preferably not less than 40–50 Micron and no bigger than 15 L); a double bag is required if a normal bag is used.<br>• The waste can be stored in a secured and locked room which must be protected from the sun, rain, water, rodents, food, and other staff.<br>• Labels of the vials and vial caps (with the aluminum seal) should be removed.<br>• Used vials should be disinfected with 0.5 ml chlorine solution (any compound containing chlorine, for example, HTH, bleach powder, NaDCC, etc.).<br>• In chlorine solution, used vials should be submerged for at least 30 minutes. After 30 minutes, vials should be removed from the solution. In both cases wearing heavy-duty gloves is a must.<br>• Chlorine solution (0.5ml) must be discarded in a safe and proper manner. This used solution can be discharged into toilets/latrines or can remain in sunlight for several hours. In any circumstances, this solution must not come into any contact with water or food.<br>• If autoclave is available, then the vials can be treated with that.<br>• After that, vials should be disposed of in a dedicated ash pit, or vials can be encapsulated so that these vials can't be reused. | • The vial caps should be kept in a separate plastic bag, and after that, these caps should be incinerated separately. |
| | Centers for Disease Control and Prevention [31] | • Should be disposed of following local regulation and process currently used to dispose of regulated medical waste | • No separate guideline | • No separate guideline |
| United States' Waste Management Agency | Stericycle [13] | • Used Syringe must be collected in a disposable or reusable sharp container<br>• The Syringe and sharps should be treated by either autoclaving or incineration. | • Fully emptied vials are recommended to collect in sharp containers to reduce probable diversion and illicit intent.<br>• Then, the empty vials should be treated by either autoclaving or incineration.<br>• The residuals dosed vials (partially emptied) are considered biohazardous medical waste/Non-hazardous pharmaceuticals. So, they should be labeled separately and treated with local regulations. | • Other waste can be collected in regular trash managed as regulated medical waste by local regulation. |
| | Biomedical Waste Services [11] | • Syringe and sharps should be collected in a US. Food and Drug Administration- approved sharp container.<br>• Then, the sharps should be disposed of following the local/CDC/ Manufacturer's guidelines. | • Same disposal method as Syringe and sharps. But vials should be collected in separate containers. | • Other hazardous materials should be collected in a red-colored bag and transported to the disposal agency. |

**Table 2.** (Continued)

| Guidelines and References | | Vaccine related Wastages | | |
|---|---|---|---|---|
| Government of India Ministry of Environment, forest and Climate Change. | [32, 33] | • Autoclaving or Dry Heat Sterilization followed by shredding or mutilation or encapsulation in a metal container or cement concrete, or a combination of shredding and autoclaving; and sent for final disposal to iron foundries (having consent to operate from the State Pollution Control Boards or Pollution Control Committees) or sanitary landfill or designated concrete waste sharp pit can be used for sharps and syringes treatment. | • Disinfection (by immersing the washed glass waste in Sodium Hypochlorite solution after cleaning with detergent and Sodium Hypochlorite treatment) or autoclaving, microwaving, or hydroplaning, and then recycling. | • Following autoclaving, microwaving, or hydroclaving, shredding or mutilation, or a combination of sterilization and shredding, treated waste should be transferred to registered or authorized recyclers, or for energy recovery, or for the conversion of plastics to diesel or fuel oil, or for road construction, if it is practicable. |
| The policy of the Bangladesh Government | [15] | • Used Syringe and related parts, e.g., needle cap and plunger cap, must be kept in a safety box.<br>• When the safety box becomes filled with three fourth portions, it should be closed and taken to a secured place.<br>• All safety boxes should be collected in the Upazila health complex or city corporation.<br>• Safety box should be burned in a pit, and the ash should be buried.<br>• Waste should be burned in an incinerator if it is available. | • Vials should be collected in a waste removal bag after the vaccination session.<br>• Used/unused, expired vials/ damaged vaccine vials should be kept in a jute sack/bag and put inside a drum. After that, these vials should be crushed by a hammer.<br>• Crushed vials should be submerged with 0.5% chlorine solution and kept for at least 30 minutes.<br>• Finally, the crushed vial bags should be removed from the drum and buried in a pit.<br>• Rest of the chlorine solution should also be poured into the pit. The pit depth must be at least 3 feet, and the pit should be surrounded by a fence.<br>• For handling chlorine solution, workers must have to wear gloves. | • Other waste should be collected in a waste removal bag and burned into a pit after transporting to the Upazila health complex or the city corporation. |

When the storage capacity of the room or the open place becomes full, the designated MT (EPI) applies for permission to treat the waste in the respective procedure to the hospital authority. The permission granting body is headed by an Upazila Health and Family Planning Officer (UHFPO). After granting permission, the health complexes treated syringes and vials separately. In the case of Syringe and sharps, most of the health complex (total 7 out of 15, and it covers 46%) burn the syringes and sharps in an open place. However, the partially burned ash was not buried under land in those hospitals. The study ranked the practices of Syringe, and sharp wastes management in the studied health complexes given in Table 4. The ranking was based on WHO's Standard Operating Procedure of COVID-19 vaccine waste management [12].

The vials were treated in a different ways in different health complex. Most of the hospitals didn't care about the government waste management policy. 53.3% of the studied health complex disinfected the vials with a 5% chlorine solution. The chlorine solution was applied to the crushed vials into a plastic drum. The vials were crushed in a plastic sack with a hammer. However, 33.33% of the health complexes didn't crush the vials. Here is, a stark contrast between the policy of WHO and Bangladesh was identified. The WHO recommended non-crushed landfilling, whereas the policy of the Bangladesh government is crushed-landfilling. However, only 37.5% of the waste management staff used gloves during disinfection. None of them used heavy-duty gloves recommended by WHO. The disposal pattern of vials is represented in (Table 5). The ranking was based on the recommendation of WHO.

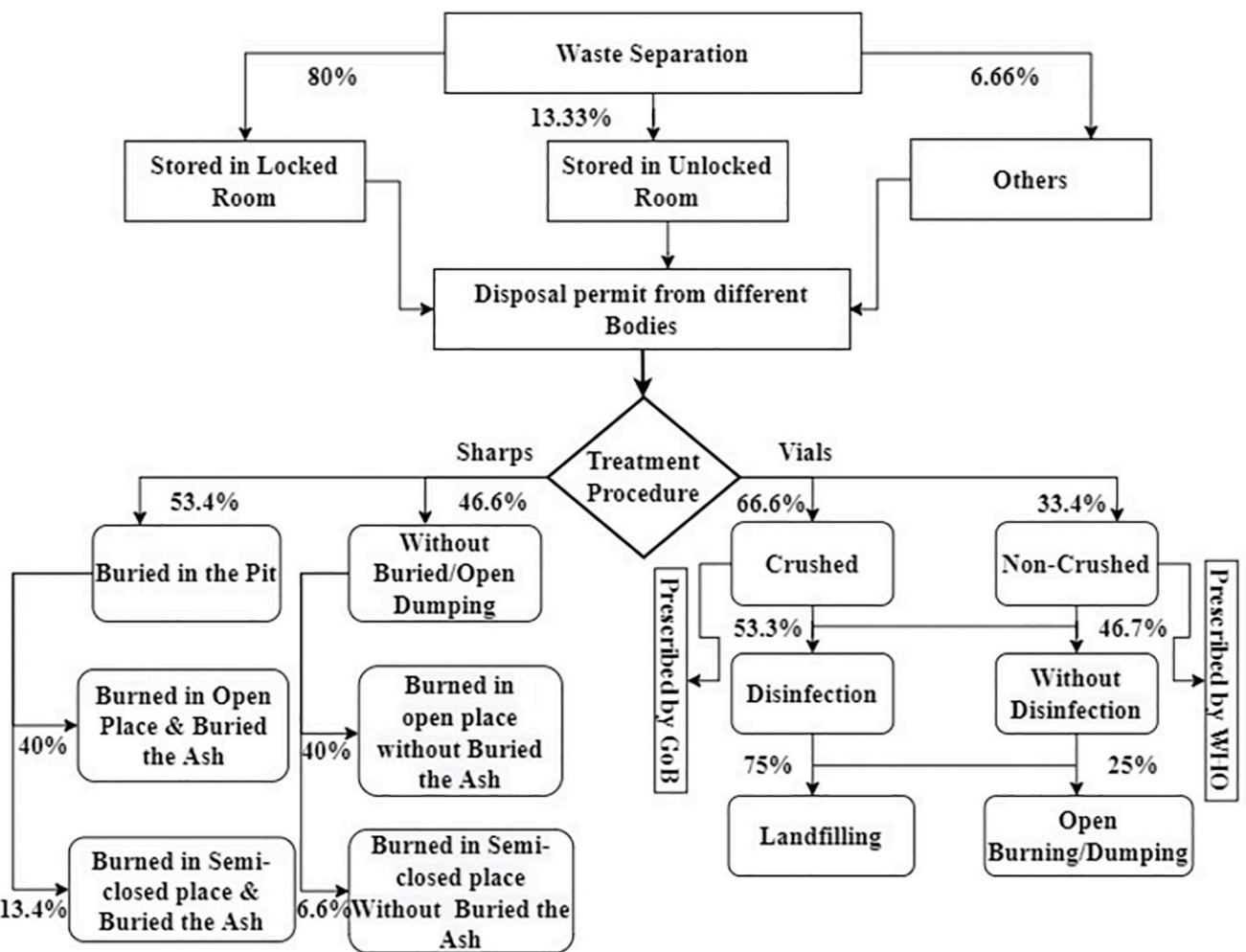

**Fig 2. Flowchart of existing COVID-19 vaccine waste management in the studied area (% in the figure denotes the number of hospital practicing the management procedure).**

**Table 3. Status of the vaccine-related waste management in the studied health complex.**

| Vaccine Waste Management Practices | Percentage of Health Complex Following Positive Practice (WHO Recommended) in % | Comments on the Status of the Investigated 4 Districts. |
|---|---|---|
| Vaccine Waste Segregation | 100 | Excellent |
| Handling of the waste | 35.5 | Very Poor |
| Safety Measures in Management activities | 46.6 | Poor |
| Storage Facility | 76 | Good |
| Disinfection status | 53.3 | Poor |
| Policy Management | 40 | Poor |

[***Excellent** (85–100%), **Good** (70–84%), **Fair** (55–69%), **Poor** (40–54%), **Very Poor** (<40%)]

**Table 4. Treatment methods for syringes and sharps disposal in the studied health complexes.**

| Treatment Methods | Comments on Treatment Method (Based on WHO) | Number of Hospitals Carrying out Treatment Method | Total Percentage of Hospitals Carrying out Treatment Method | Remarks |
|---|---|---|---|---|
| Autoclave and Incineration | Excellent | 0 | 0 | Not Practiced |
| Incineration | Good | 0 | 0 | Not Practiced |
| Semi-Close Burning and Buried subsequently | Fair | 2 | 13.33 | Rarely Practiced |
| Open Burning and Buried subsequently | Poor | 6 | 40 | Often Practiced |
| Open/semi-close Burning without Buried | Very Poor | 7 | 46.66 | Mostly Practiced |

[The scaling of 'Comments' was constructed on the basis of Standard Operating Procedures (SOP) of WHO [12]]

## 3.3 Waste generation estimation

Till 12-December of 2021 total number of Vaccine, including first and second doses, are $8.61 \times 10^7$. So, the same number of syringes were used for vaccination. One unit of Syringe, including the needle, needle cap, and packaging material, is 4.4557gm. Finally, the amount of waste of syringes and sharps generated till 12-December of 2021 was 576 tonnes. The total target of vaccination of the Bangladesh government is $1.38 \times 10^8$, which covers 80% of the country's total population. As a result, the total amount of estimated waste of syringes and sharps will be 1,232 tonnes (Table 6)

**Table 5. Treatment methods for vials disposal in the studied health complex.**

| Treatment Methods | Comments on Treatment Method (Based on WHO) | Number of Hospitals Carrying out Treatment Method | Total Percentage of Hospitals Carrying out Treatment Method | Remarks |
|---|---|---|---|---|
| Non-crushed encapsulation (Reuse) | Excellent | 0 | 0 | Not Practiced |
| Sterilized without cap and Non-Crushed Landfilling | Good | 1 | 6.66 | Rarely Practiced |
| Disinfection and Crushed Land Filling | Fair | 6 | 40 | Most Often Practiced |
| Without Disinfection and Crushed/Non-Crushed Land Filling | Poor | 4 | 26.66 | Often Practiced |
| Crushed or Non-Crushed Open Burning/ Open Dumping without landfilling | Very Poor | 4 | 26.66 | Often Practiced |

[The scaling of 'Comments' was constructed on the basis of Standard Operating Procedures (SOP) of WHO [12]]

**Table 6. Waste estimation of syringes and sharps (full packet).** [Data Source: [7]].

| | | |
|---|---|---|
| Weight of Syringe with Full Pack (gm) | | 4.4557 |
| Total No. Doses till 12/12/2021 | Single Dose | $4.29 \times 10^7$ |
| | Double Dose | $4.32 \times 10^7$ |
| Syringe Waste Generated till 12/12/2021 (tonnes) | | 576 |
| Syringe Waste Generated Per Month (Feb- Dec-2021) | | 58 |
| Targeted Vaccine Doses (80% of Population) | | $1.38 \times 10^8$ |
| Estimated Total Waste (tonnes) | | 1,232 |

Four types of Vaccines had given to the people of Bangladesh so far. Unfortunately, we could not be able to take any sample of Pfizer vaccine vials from any of our studied areas. Either the vials of Pfizer were disposed of or not allotted to their health complex. However, the other 3 types of vaccine vials named Moderna (10 doses), Astrazeneca (10 and 1doses), and Sinopharm (2 and 5 doses) were collected. Covishield is the Vaccine of Indian Serum Institute considered Astrazeneca in the government website as it contains the formula of Astrazeneca as well [7]. The total number of 10 doses contained Moderna vials used as per the vaccination doses was $2.98 \times 10^6$, which produced a total of 6.27 tonnes of vials waste till 12-December 2021. Like all the Moderna vaccines, Astrazeneca and Sinopharm were not mono-numbered vials. As we were not able to obtain the exact number of vials imported into Bangladesh. So, we made an estimation of the probable waste generation from each vial.

The total number of Astrazeneca vaccine doses given to date was $2.0 \times 10^7$. If we consider all the vials were 10 doses, the total waste generation would be 25.03 tonnes. Considering all vials of Astrazeneca were single, the total waste generation would be 98.40 tonnes. By assessing the difference in weight of single and 10 doses vials, it can be concluded that increasing the doses in a single vial would be reduced the generation of waste. Ten doses containing Covishield, an Astrazeneca formulized Vaccine provided by Serum Institute of India, possess a lower weight than Astrazeneca 10 doses. If we assume the total AstraZeneca vaccine was Covishield, then the waste generation to date would be 13.39 tonnes which contains almost half of the weight of the Astrazeneca 10 doses. It can be concluded that Covishield produces less amount of waste than the 10 doses containing Astrazeneca.

In the case of Sinopharm, if all the given vaccine vials contained 5 and 2 doses, 121.07 tonnes and 257.85 tonnes of waste would be generated by 12-December-2021, respectively. The fact that waste increases with the decrease of doses per vial were also proved by comparing the weight of 2 and 5 doses containing vials of Sinopharm. All the vials related waste generation estimations are represented as follows (Table 7).

The wastes generated from syringes, sharps, and vials will surely impose an extra burden on existing waste management practices. Previously $3.18 \times 10^3$ tonnes of medical waste was generated per month due to COVID-19 in Dhaka. The vaccination program produced 58 tonnes of syringes and sharps waste per month and added to the current medical wastes. It also imposed a huge burden of waste per month produced from vaccine vials. Thus to achieve the targeted vaccination, the existing poor medical waste management facility in Bangladesh will face an extreme challenge to manage the extra waste burden from vaccination programs.

## 3.4 WOT analysis as a part of strategic planning

In Fig 3, an overview of the SWOT analysis is presented.

**Table 7. Estimation of already generated COVID-19 vaccine waste by the vials of different brands.** [Data Source: [7]].

| Name of Vaccine Brand | Moderna | Astrazeneca | | | Sinopharm | |
| --- | --- | --- | --- | --- | --- | --- |
| | | | | Covishield | | |
| Doses Containing a Single Vial | 10 | 10 | 1 | 10 | 5 | 2 |
| Weight of the Vials (gm) | 11.7243 | 12.519 | 4.9214 | 6.6988 | 6.0567 | 5.1599 |
| Completion of First Doses | $2.72 \times 10^6$ | $1.30 \times 10^7$ | $1.30 \times 10^7$ | $1.30 \times 10^7$ | $6.75 \times 10^7$ | $6.75 \times 10^7$ |
| Completion of Second Doses | $2.63 \times 10^6$ | $7.00 \times 10^6$ | $7.00 \times 10^6$ | $7.00 \times 10^6$ | $3.25 \times 10^7$ | $3.25 \times 10^7$ |
| Estimated Total Generated Vials Waste (Tonnes) | 6.27 | 25.03 | 98.40 | 13.39 | 121.07 | 257.85 |

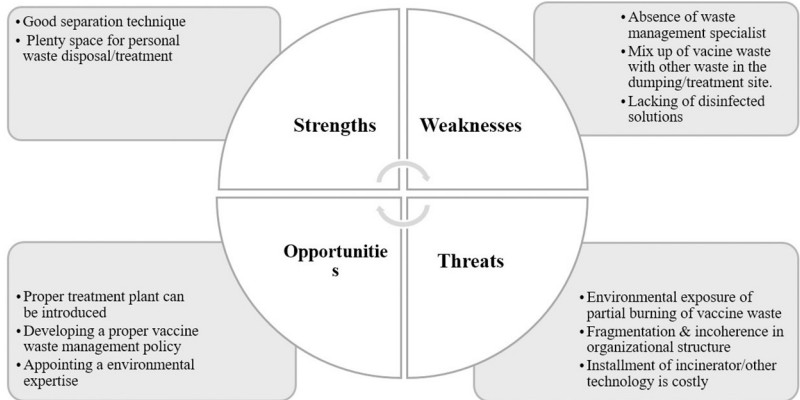

**Fig 3. SWOT analysis for COVID-19 vaccine waste management in the investigated.**

M. Shammi et al. 2022 previously studied the SWOT analysis of Bangladesh's Biomedical Waste Management facilities [34]. The present study investigated the SWOT analysis solely of the COVID-19 vaccine waste management system at the Upazila level of Bangladesh.

The current management system for the COVID-19 vaccine waste has improved than the previous EPI vaccine waste management system but still has a gap in this management practice. The internal strength found in the studied hospital is that they segregated the sharps and vials wastage in a proper way. They put the sharps and syringes in a safety box that the Government supplied. All the investigated hospitals disposed of/treated their wastage in their personal place, and the place was large enough to support the procedure. EPI medical technician was given responsibility for the vaccine wastage disposal. As they managed the vaccine wastage of the EPI program, they are quite experienced in this field and have moderate knowledge about the management procedure. The wastage was kept in a separate room, and in the case of the maximum hospital, the room was protected from the sun, rodents, rainwater, etc. In every stage of the management procedure, the associated staff wore masks to protect themselves.

The following table (Table 8) shows the internal strength and weakness of the investigated hospitals:

The staff of these hospitals didn't get proper training in wastage management. They received online training in which waste management was a small part. There was no specialist for this management; EPI medical technician did this as a part of their job. As a result, the management was not done properly. Wastage was mixed with other waste at the dumping site, and anyone could access it. The storing materials for vials after vaccination was poor, and some hospital authorities put them in an open packet, box, or sack. The amount of disinfection solutions provided by the Government was not enough, and the Government didn't provide heavy-duty gloves for handling this waste, as the MT(EPI) said in the interview. Overall, there was no separate waste management system for handling this huge amount of waste which created a burden.

As every hospital has enough space, a treatment plant can be installed at the Upazila level. If an incinerator is installed on the hospital premises, then every other private hospital can centrally manage this waste in the treatment plant, and as a result, the cost of the treatment will be reduced. Vaccination is a regular program, and for different issues, people need vaccination so that Government can produce a separate management procedure for the management of Vaccine related waste. The person who was appointed has a moderate knowledge of management.

**Table 8. SWOT analysis for COVID-19 vaccine waste management (strengths and weaknesses).**

| Strength | Weaknesses |
|---|---|
| 1. Good separation technique for the management of COVID-19 vaccine-related waste | 1. Staff didn't receive proper training on the COVID-19 vaccine waste management. It was only a small part of the vaccination program. |
| 2. Separate safety box for sharps and Syringe | 2. Absence of waste management specialist. |
| 3. Enough space for personal waste disposal/treatment | 3. Poor packaging materials for vials and other wastage except for sharps and syringes. |
| 4. Separate storeroom for the collection of the waste. | 4. The public can access the waste dumping site. |
| 5. Experienced staff in the management of EPI vaccine waste. | 5. Although vaccine waste was separated in the hospital, they were treated/disposed of with other waste at the disposal site. |
| 6. The staffs have moderate knowledge about the vaccine waste management. | 6. Poor handling of this wastage (e.g., The authorized person didn't use gloves, PPE, etc.) |
| 7. Wear mask in every phase of waste management. | 7. Lacking of heavy-duty gloves for disinfection. |
| | 8. Lacking of disinfected solutions. |
| | 9. Didn't separate vaccine cap during the treatment. |
| | 10. Absence of autoclave for disinfection. |
| | 11. Open and partial burning of vaccine waste. |
| | 12. Land disposal of vials waste with rare pretreatment. |
| | 13. Poor disposal of chlorine-related compounds. |
| | 14. Rarely had good pit systems. |
| | 15. Absence of separate waste management practice for COVID-19 vaccine-related waste. |

Through proper training and guidelines, they can be used productively. These are the external opportunities to improve the current management procedure.

The following table (Table 9) shows the opportunities and threats of the investigated hospital:

Being a developing country, the management procedure has to face some threats. As the mass vaccination program started and the country has a vast population, a huge amount of

**Table 9. SWOT analysis for COVID-19 vaccine waste management (opportunities and threats).**

| Opportunities | Threats |
|---|---|
| 1. Proper treatment plant can be introduced at Upazila level. | 1. Environmental exposure of partial burning of vaccine waste |
| 2. Environmental expertise can be appointed for each hospital. | 2. A huge amount of COVID-19 vaccine waste is produced in a short time. |
| 3. Skill improvement of the responsible authority by providing proper training. | 3. Scarcity of fund for proper treatment |
| 4. Creating adequate fund and research. | 4. Lack of knowledge about impacts of partial burning |
| 5. Government can develop a proper vaccine waste management policy. | 5. Disease outbreak due to improper managements. |
| 6. Proper technological improvement can be introduced. | 6. Installment of incinerator/other technology is costly |
| 7. Autoclave and other disinfection materials can be provided. | 7. Not having qualified staff |
| 8. Safety materials, including heavy-duty gloves, boots, PPE, masks etc., can be provided by the Government/NGO. | 8. Regional and global economic crisis |
| | 9. The need of reducing the budget deficit. |
| | 10. Fragmentation and incoherence in organizational structure. |
| | 11. Current management is not up to the international standard. |
| | 12. Misuse and Mismanagement of existing Autoclaves may lead to further pollution and health burdens. |

COVID-19 vaccine waste has been produced in a short time. It creates a scarcity of funds for the proper treatment of wastage.

### 3.5 DPSIR framework

We applied the DPSIR Framework for the management procedure of COVID-19 vaccine wastage in the 15 Upazilas of 4 Districts. Fig 4 shows the framework as a whole.

**3.5.1 Driving force for the management of COVID-19 vaccine waste.** The 7th human coronavirus, severe acute respiratory syndrome coronavirus 2 (SARS-CoV- 2), was found in Wuhan, Hubei Province, China, throughout a current pneumonia outbreak in January 2020 [35]. The COVID-19 pandemic has had quite a major influence on society, with governments throughout the world establishing travel restrictions and other measures to keep the virus from spreading, such as forced face covers or quarantine [36]. COVID-19 pandemic has also significantly impacted health (disease burden). Multiple organs are frequently affected in severely unwell individuals. The virus binds to ACE2 receptors in vascular endothelial cells, the heart, the brain, the kidneys, the colon, the liver, the pharynx, and other tissues. It has the potential to harm these organs directly. In addition, the virus's systemic problems induce organ dysfunction [37]. There is no specific treatment procedure for the treatment of this disease. Vaccines against COVID-19 are believed to be extremely important for preventing and controlling COVID-19 since immunization is one of the most effective and cost-efficient health strategies for preventing infectious illnesses [38].

Bangladesh reported its first case in March 2019; after that, the number of COVID-19 cases increased day by day. As a result, the hospitals cannot cope with the cases, and the importance of vaccination programs is getting higher. Following this trend, the first vaccination program was started on 27th January 2021.

**3 5.2 Pressures and states for the management of COVID-19 vaccine waste.** The hospital waste management system is already poor in Bangladesh. Though medical waste possesses only one percent of solid waste in the country, it is not managed properly. In maximum cases,

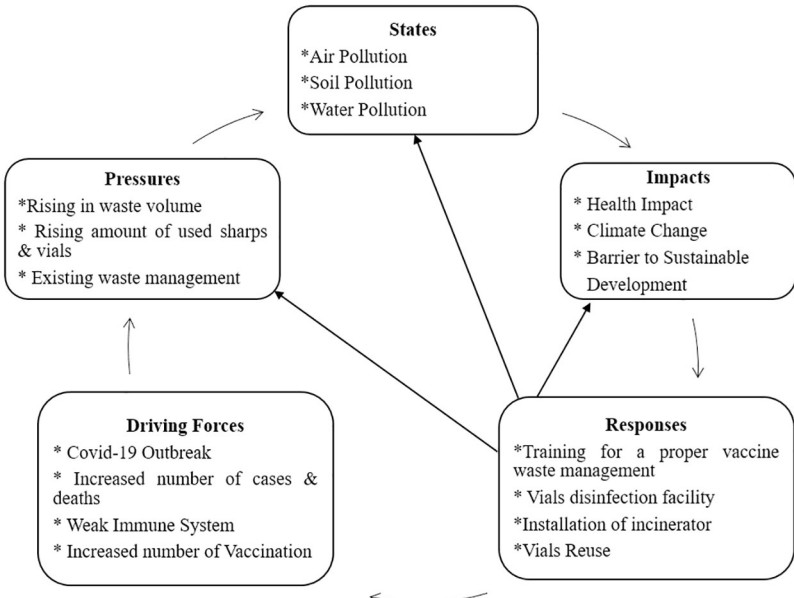

**Fig 4. DPSIR framework of the studied area.**

they are mixed with other household wastage [39]. Globally, an estimated 41% of garbage is burnt publicly. This figure is significantly greater in developing, low-income nations. Approximately 620 million tons of garbage are burnt each year openly. Significant volumes of greenhouse gases are released into the environment when open garbage is burned [40]. Due to the COVID-19 mass vaccination program, a huge amount of waste is produced at a time in the corresponded hospital in the country. This huge amount of vaccine wastage (rising sharps, vials, packaging materials) has created an extra burden upon the existing hospital waste management system. This pressure (Vaccine related wastages) has created the state of the environment that effected the quality of the environmental components e.g., air, water and soil. As the sharps and syringes burned in an open system it created the air pollution. Improper disinfection of vials and buried into the soil had created soil pollution and many hospitals has the disposal site besides a water body which caused harm in the quality of the water.

**3.5.3 Impacts and responses for the management of COVID-19 vaccine waste.** Carbon dioxide, methane, and particulate matter are examples of substances that are commonly connected with air pollution and can cause severe respiratory disease. Open waste burning is particularly linked to the release of persistent organic pollutants. This includes carcinogens such as polycyclic aromatic hydrocarbons, dioxins, and furans, which have all been linked to a range of ailments. Children and unborn fetuses are also vulnerable to exposure to pollutants. High exposure to $CO_2$ in the air due to open burning may increase blood pressure, headache, heart and respiratory rate, dizziness, lung cancer, asthma, etc. The gases released from the open burning also contributed to changes in regional and global climate [40]. [41] stated that the mismanagement of wastage, including open burning a serious barrier to achieving sustainable development. One point of global waste management goals for improving sustainability is to stop uncontrolled dumping and open burning globally [42]. Waste generated from medical is more high potential for disease transmission than other waste. When medical waste is burned in an open fire, it generates hazardous fumes pollute the atmosphere. If preventative precautions are not followed, these emissions might cause respiratory and skin problems, as well as cancer [43]. Partial burning of medical waste eventually releases toxic gas, which has several health impacts. It also creates ashes causing secondary handling and treatment problems [44].

With the increasing amount of vaccine waste and its impact on the environment, it is obvious to take some necessary steps. Proper training for the management of COVID-19 vaccine waste is an essential part of minimizing the detrimental impact. Disinfection of vials can solve the problem of soil pollution, and proper discharge measure for chlorine-related disinfection solution is necessary to mitigate environmental impact. Vials can be encapsulated, and the use of the same vials can be minimized through this encapsulation. Encapsulation is a process through which vials are mixing them within a cement, lime, and water mixture (3/3/1 parts by weight) in a sealed metallic drum. A central incinerator can be arranged in an Upazila, and all the hospital in this Upazila can treat their waste in the central incinerator.

## 4. Conclusions

The study finds the current trends in the management of the COVID-19 vaccination program in Bangladesh. The investigation discovered deficiencies and inefficiencies in the current waste management system, which could exacerbate Vaccine-waste mishandling and leakage into the environment, resulting in a new environmental crisis. Studied Upazila health complex had enough space to build up a proper treatment plant in their own complex. If it is subjected to be an expensive plan, at least a central treatment plant of biomedical waste for several neighbor Upazila and a small-sized autoclave for each Upazila hospital can be provided to sterilize

the vials waste. The study had limitations in exact data acquisition for total vials waste estimation. The present analysis suggests doing life cycle assessments (LCA) of the COVID-19 vaccine vials, syringes, and sharps. It also proposes testing the efficiency of reusing the vaccine vials after encapsulation as a substitute for concrete in making river-side dams or other uses.

## Acknowledgments

We would like to thank the medical technologist and other staff for sharing the valuable information. We also want to acknowledge the contribution of Rehnuma Karim Rinky for giving fruitful guidelines to prepare the questionnaire.

## Author Contributions

**Conceptualization:** Md Rayhanul Islam Rayhan, Jannatul Mawya Liza, Md. Mostafizur Rahman.

**Data curation:** Md Rayhanul Islam Rayhan, Jannatul Mawya Liza, Md. Mostafizur Rahman.

**Formal analysis:** Md Rayhanul Islam Rayhan, Jannatul Mawya Liza, Md. Mostafizur Rahman.

**Investigation:** Md Rayhanul Islam Rayhan, Jannatul Mawya Liza, Md. Mostafizur Rahman.

**Methodology:** Md Rayhanul Islam Rayhan, Md. Mostafizur Rahman.

**Project administration:** Md. Mostafizur Rahman.

**Supervision:** Md. Mostafizur Rahman.

**Validation:** Md. Mostafizur Rahman.

**Writing – original draft:** Md Rayhanul Islam Rayhan, Jannatul Mawya Liza.

**Writing – review & editing:** Md. Mostafizur Rahman.

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
