## [Decision Letter · Decision Letter 0]

4 Apr 2022

PONE-D-21-41006COVID-19 vaccination related medical waste; quantification and management perspectives in BangladeshPLOS ONE

Dear Dr. Rahman,

Thank you for submitting your manuscript to PLOS ONE. After careful consideration, we feel that it has merit but does not fully meet PLOS ONE’s publication criteria as it currently stands. Therefore, we invite you to submit a revised version of the manuscript that addresses the points raised during the review process.

We look forward to receiving your revised manuscript.

Kind regards,

Nirupam Aich

Academic Editor

PLOS ONE

Journal Requirements:

3. Please remove your figures from within your manuscript file, leaving only the individual TIFF/EPS image files, uploaded separately.  These will be automatically included in the reviewers’ PDF

4. We note that Figures 2.1  in your submission contain [map/satellite] images which may be copyrighted. All PLOS content is published under the Creative Commons Attribution License (CC BY 4.0), which means that the manuscript, images, and Supporting Information files will be freely available online, and any third party is permitted to access, download, copy, distribute, and use these materials in any way, even commercially, with proper attribution. For these reasons, we cannot publish previously copyrighted maps or satellite images created using proprietary data, such as Google software (Google Maps, Street View, and Earth). For more information, see our copyright guidelines: http://journals.plos.org/plosone/s/licenses-and-copyright.

   a. You may seek permission from the original copyright holder of Figures 2.1  to publish the content specifically under the CC BY 4.0 license. 

Reviewers' comments:

Reviewer's Responses to Questions

**Comments to the Author**

1. Is the manuscript technically sound, and do the data support the conclusions?

Reviewer #1: Partly

2. Has the statistical analysis been performed appropriately and rigorously? 

Reviewer #1: No

3. Have the authors made all data underlying the findings in their manuscript fully available?

Reviewer #1: No

4. Is the manuscript presented in an intelligible fashion and written in standard English?

Reviewer #1: No

5. Review Comments to the Author

Reviewer #1: The study is a timely and important one to help prescribe and implement guidelines for vaccine related medical waste management in countries where waste management practices are already poor. The data collection is sound. However however data analysis and methods of calculation of some of the major parameters are unclear and missing. Some points of discussion are missing and have been detailed below.

Due to grammatical errors throughout the manuscript, the readability is poor. It is highly recommended to address the grammatical errors by the authors to make the scientific information more readable.

Major comments:

Introduction section:

1. The objective of exploring and comparing the guidelines of various organizations for COVID-19 vaccination related waste disposal is not mentioned explicitly in the Introduction section. (It is present in the abstract, but further details are missing in the Introduction and Methods section)

2. The choice of these particular 15 upazilas in these particular 4 districts is unclear. Why were these particular upazilas chosen? It would be good for the international reader to know the definition of upazilas and the brief hierarchy of upazilas and districts, if possible.

3. As it might be a preliminary study, the number of upazilas and districts chosen are sufficient. However, in other districts and upazilas, is there any evidence of the situation being, better, similar or worse than described in this study? This can be discussed using known evidence, if available. Alternatively, upazila-wise comparison of the waste management practices would shed light on whether this is the average situation, or are there some outliers which are carrying out waste management steps which are much below standard. Including case by case comparison can help shed light on the region/country.

Methods and Materials:

4. Section 2.4: It is not mentioned, how many data points (was the average of several weights of vials taken, if yes, how many data points were taken. Also the error interval must then be mentioned).

5. Is the summation symbol misplaced in equations (i) to (iv)? Aren’t Sw and Swm the total weight and total weight per month ? Closing brackets are missing for Equation (i) and some terms seem to be missing in equation (ii). In equation (iv) why isn’t N2 multiplied by 2.

Results and Discussion

6. Section 3.1 : This objective has not been written in the Introduction section (See comments for Introduction section). Also the methodology (even if brief) related to Section 3.1 missing from ‘Methods and Materials’ section.

7. Figure 3.1: The flowchart can be improved for higher clarity of the results. It would be good to see what percentage of waste is being disposed of in which method, and whether the method used comes under prescribed guidelines. For eg. ‘Crushed’ and 'Not crushed' can go to two different flow paths showing the percentage, indicating the better ‘prescribed pathway’.

8. Results and Discussion: The SWOT analysis does not include any references to similar countries/regions facing the medical waste management issues similar to this study.

Minor comments:

1. Line 48-49: The vaccination names mentioned are not categories of vaccine, rather they are the manufacturer or brand names. Categories can include mRNA vaccine or viral vector vaccines. This needs to be clarified.

2. Line 53: Add references to the fact that medical waste management system being damaged in Bangladesh

3. Line 67: "when turning into wastage required an intense energy." Elaborate how waste management of glass requires intense energy with proper references

4. Line 85: What is an “EPI”? Please define it.

5. Line 92: Add citations related to “Previously a lot of research work”

6. Line 114: The questionnaires were directed to “Medical Technologist of EPI (MT-EPI) or/and EPI Superintendent" . It is unclear whether they were in charge of the whole upazila or various health complexes of the upazila?

7. Table 3.1: Column 2 “Labels and vial caps”?

8. Figure 3.1: “Different permits from different bodies”. What are these bodies? Can it be cleared in the methods or discussion section?

9. Line 238: “Considering all of these factors, the study found 76% of good practices in the studied health complex which scaled as a good practice” Can the authors clarify the calculation of the 76%. The calculation of percentages in Table 3.2 is unclear because not all the questions of the questionnaire in Section 2.3 are of ‘Yes/No’ type. The number of complexes showing ‘positive practices’ are chosen on which criteria, could the authors clarify? The calculation of all the percentages can be shown in the ‘Methods and materials’ section.

10. Line 128: ‘Paper-made carton” ?

11. Line 242-254: Why were inverted commas/quotation marks used for these lines? Isn’t it part of the authors’ manuscript or is it from another source?

12. Line 258: What does (7, 46%) mean?

13. Line 260: “( Table 3.3 )" Why brackets have been used to refer to this Table?

14. Table 3.2: Column 2 could be renamed ‘Percentage of health complexes following good practice (WHO Recommended) (%)’ for clarity. In column 3, is the ‘status’ for the four districts studied or for whole of Bangladesh? Can be clarified in the column heading or caption

15. Tables 3.3 and 3.4: Do the column headings mean 'Number of Hospitals carrying out treatment method' and 'Total percentage of hospitals carrying out treatment method' ?

16. It is recommended to shift the column 'Comments on treatment methods; (BASED on WHO)’ as the second column of this table for better understanding. Also further clarification of the calculation of percentage of this table is required. (See minor comment 14)

17. Line 299: The method of estimation should be in the 'METHODS' section.

18. Table 3.6: Is the waste generation estimation the total one that will be generated after all doses are completed? Or just which have been generated so far? The heading should be clarified modified accordingly.

19. Line 348: Is there any data as to the gap of disinfection material in kg/ tonne/ percentage? If yes, it should be mentioned quantitatively, and cited if required.

20. SWOT analysis: Among other threats, the authors can consider autoclave misuse and mismanagement leading to further pollution and health burden

General grammatical or syntactical errors:

• Use of ‘covid’ or ‘COVID’ not uniform.

• Line 43: What does 'induced' mean? Do the authors mean to say 'infection due to'

• Line 47: Usage of 'cut above the rest' inappopriate in this sentence.

• Line 107: 'Bangladesh governments'? The phrase is unclear as to which entities the suggestions are coming from.

6. PLOS authors have the option to publish the peer review history of their article (what does this mean?). If published, this will include your full peer review and any attached files.

Reviewer #1: **Yes: **Nafisa Islam

---

## [Author Response · Author response to Decision Letter 0]

21 May 2022

Response to the Reviewer`s Comments: 

We thank editor and reviewer for their positive feedback on our manuscript. We have revised the manuscript considering all the issues raised by the reviewer and the changes are shown in the track change and highlighted text in the main manuscript and other figure and table files. A point-by point response has been given as follows: 

Reviewer #1: The study is a timely and important one to help prescribe and implement guidelines for vaccine related medical waste management in countries where waste management practices are already poor. The data collection is sound. However, data analysis and methods of calculation of some of the major parameters are unclear and missing. Some points of discussion are missing and have been detailed below.

Response: Authors would like to thank reviewer for positive feedback on the manuscript along with the excellent suggestions for the improvement. We have revised the entire manuscript following the reviewers` comments. 

Due to grammatical errors throughout the manuscript, the readability is poor. It is highly recommended to address the grammatical errors by the authors to make the scientific information more readable.

Response: Thank you for your observation. As we are not from the English native speaking nations but we have tried to improve the grammatical errors during revision. 

Major comments:

Introduction section:

1. The objective of exploring and comparing the guidelines of various organizations for COVID-19 vaccination related waste disposal is not mentioned explicitly in the Introduction section. (It is present in the abstract, but further details are missing in the Introduction and Methods section)

Response: The objective of exploring and comparing the guidelines of various organizations for COVID-19 vaccine-related waste disposal has been updated in the introduction section (Line 90 to 96). Thank you!

2. The choice of these particular 15 upazilas in these particular 4 districts is unclear. Why were these particular upazilas chosen? It would be good for the international reader to know the definition of upazilas and the brief hierarchy of upazilas and districts, if possible.

Response: The reason for choosing 15 Upazilas of four districts and the definition of Upazilas as well as a hierarchy has been mentioned in the introduction section. Thank You!

3. As it might be a preliminary study, the number of upazilas and districts chosen are sufficient. However, in other districts and upazilas, is there any evidence of the situation being, better, similar or worse than described in this study? This can be discussed using known evidence, if available. Alternatively, upazila-wise comparison of the waste management practices would shed light on whether this is the average situation, or are there some outliers which are carrying out waste management steps which are much below standard. Including case by case comparison can help shed light on the region/country.

Response: As we have mentioned in the manuscript, there are 495 Upazilas in the country, and they are following the same guideline for the treatment of vaccine waste. So, there is little chance that their management would be better than these investigated Upazilas. However, we do agree that some of the upazilas near might get treatment facility provided by the PRISM Bangladesh foundation but the thing is that their capacity is very limited to deal with the incremental COVID-19 vaccination related wastes. Thus, the scenario is more of less similar all over the country. 

Methods and Materials:

4. Section 2.4: It is not mentioned, how many data points (was the average of several weights of vials taken, if yes, how many data points were taken. Also the error interval must then be mentioned).

Response: The weight of the empty vial is the same for each brand. Thus we just collect a single vial for each brand as a sample. Also, the weight of the Syringe & associated packaging material is the same all over the country. This information is now included in the manuscript. Thank you!

5. Is the summation symbol misplaced in equations (i) to (iv)? Aren't Sw and Swm the total weight and total weight per month? Closing brackets are missing for Equation (i) and some terms seem to be missing in equation (ii). In equation (iv) why isn't N2 multiplied by 2.

Response: We apologize for the mistake of wrongly placing the summation symbol. The summation symbol is corrected in the equations (i) to (iv). The issue regarding Sw & Swm is fixed. The closing bracket in equation (i) is placed. The term Nd was misplaced; now, it is fixed up. In equation (iv), N2 wasn't multiplied by 2 as this equation calculated the waste generated from vaccine vials of the only second dose (Not the sum of the first & second dose). However, it is now more specified in the manuscript. Thank you!

Results and Discussion

6. Section 3.1 : This objective has not been written in the Introduction section (See comments for Introduction section). Also the methodology (even if brief) related to Section 3.1 missing from 'Methods and Materials' section.

Response: The objective has been updated in the introduction section. The methods for section 3.1 are described in section 2.2 of the methodology section (Secondary Data Analysis). However, section 2.2 has also been updated slightly. Thank you!

7. Figure 3.1: The flowchart can be improved for higher clarity of the results. It would be good to see what percentage of waste is being disposed of in which method, and whether the method used comes under prescribed guidelines. For eg. 'Crushed' and 'Not crushed' can go to two different flow paths showing the percentage, indicating the better 'prescribed pathway'.

Response: The flowchart has been updated as per your suggestion. Thank you!

8. Results and Discussion: The SWOT analysis does not include any references to similar countries/regions facing the medical waste management issues similar to this study.

Response: Reference has been added as per your suggestion. Thank you!

Minor comments:

1. Line 48-49: The vaccination names mentioned are not categories of vaccine, rather they are the manufacturer or brand names. Categories can include mRNA vaccine or viral vector vaccines. This needs to be clarified.

Response: The classification as per your suggestion has been updated. Thank you!

2. Line 53: Add references to the fact that medical waste management system being damaged in Bangladesh

Response: Reference has been added. Thank you!

3. Line 67: "when turning into wastage required an intense energy." Elaborate how waste management of glass requires intense energy with proper references

Response: The energy required to manage glass waste has been discussed elaborately with references. Thank you!

4. Line 85: What is an "EPI"? Please define it.

Response: Elaborate form of EPI has been added. Thank you!

5. Line 92: Add citations related to "Previously a lot of research work"

Response: The Citation has been added. Thank you!

6. Line 114: The questionnaires were directed to "Medical Technologist of EPI (MT-EPI) or/and EPI Superintendent" . It is unclear whether they were in charge of the whole upazila or various health complexes of the upazila?

Response: The in-charge of Upazila Health Complex has been clearly demonstrated. Thank you!

7. Table 3.1: Column 2 "Labels and vial caps"?

Response: Sorry! This was a spelling mistake! The spelling mistake has been corrected (Levels to Labels). Thank you!

8. Figure 3.1: "Different permits from different bodies". What are these bodies? Can it be cleared in the methods or discussion section?

Response: The bodies incorporated with the permission procedure are added to the manuscript (In the discussion part, Line no 297). Thank you!

9. Line 238: "Considering all of these factors, the study found 76% of good practices in the studied health complex which scaled as a good practice" Can the authors clarify the calculation of the 76%. The calculation of percentages in Table 3.2 is unclear because not all the questions of the questionnaire in Section 2.3 are of 'Yes/No' type. The number of complexes showing 'positive practices' are chosen on which criteria, could the authors clarify? The calculation of all the percentages can be shown in the 'Methods and materials' section.

Response: The factors & sub-factors that indicate the positive practices, and the equation have been added in the methodology section (Section 2.4). Thank you!

10. Line 128: 'Paper-made carton"?

Response: The term has been rearranged for better understanding. Thank you!

11. Line 242-254: Why were inverted commas/quotation marks used for these lines? Isn't it part of the authors' manuscript or is it from another source?

Response: It was the description of the Table 3.2. The inverted comma has been removed. Thank you!

12. Line 258: What does (7, 46%) mean?

Response: Here, 7 means Health Complex number, and it covers 46%. This issue has been resolved in the manuscript. Thank you!

13. Line 260: "( Table 3.3 )" Why brackets have been used to refer to this Table?

Response: The brackets have been removed. Thank you!

14. Table 3.2: Column 2 could be renamed 'Percentage of health complexes following good practice (WHO Recommended) (%)' for clarity. In column 3, is the 'status' for the four districts studied or for whole of Bangladesh? Can be clarified in the column heading or caption

Response: The issues have been resolved in table 3.2's column heading. Thank you!

15. Tables 3.3 and 3.4: Do the column headings mean 'Number of Hospitals carrying out treatment method' and 'Total percentage of hospitals carrying out treatment method' ?

Response: Yes, the column headings mean number and the total percentage of hospitals carrying out treatment methods and it has been added to the column heading. Thank you! 

16. It is recommended to shift the column 'Comments on treatment methods; (BASED on WHO)' as the second column of this table for better understanding. Also further clarification of the calculation of percentage of this table is required. (See minor comment 14)

Response: The column has been shifted as per your suggestion also, some corrections are incorporated in the manuscript. Thank you!

17. Line 299: The method of estimation should be in the 'METHODS' section.

Response: The Method of waste estimation is already stated in the METHODS section. For your reference, it is the Equation no. (v) under section 2.5. Thank you!

18. Table 3.6: Is the waste generation estimation the total one that will be generated after all doses are completed? Or just which have been generated so far? The heading should be clarified modified accordingly.

Response: The heading of Table 3.6 has been updated. The element of the Table has also been updated. Thank you!

19. Line 348: Is there any data as to the gap of disinfection material in kg/ tonne/ percentage? If yes, it should be mentioned quantitatively, and cited if required.

Response: There is no such literature about this information. The statement was adopt from the responsible MT(EPI) officers. However, It has been further clarified in the manuscript. Thank you!

20. SWOT analysis: Among other threats, the authors can consider autoclave misuse and mismanagement leading to further pollution and health burden

Response: You are absolutely right; this point may also be added. However, It has been updated as per your suggestion.

General grammatical or syntactical errors:

• Use of 'covid' or 'COVID' not uniform.

Response: It has been fixed in the whole maniscript. Thank you!

• Line 43: What does 'induced' mean? Do the authors mean to say 'infection due to'

Response: It was meant exactly what you think. However, it has been clarified in the manuscripts. Thank you!

• Line 47: Usage of 'cut above the rest' inappopriate in this sentence.

Response: It has been solved. Thank you!

• Line 107: 'Bangladesh governments'? The phrase is unclear as to which entities the suggestions are coming from.

Response: The entity has been identified in the manuscript. Thank you!

Finally, again we would like thank the reviewer for excellent suggestions that definitely helps to improve the quality of the manuscript.

---

## [Decision Letter · Decision Letter 1]

2 Aug 2022

COVID-19 vaccination related medical waste; quantification and management perspectives in Bangladesh

PONE-D-21-41006R1

Dear Dr. Rahman,

We’re pleased to inform you that your manuscript has been judged scientifically suitable for publication and will be formally accepted for publication once it meets all outstanding technical requirements.

Kind regards,

Nirupam Aich

Academic Editor

PLOS ONE

Additional Editor Comments (optional):

Reviewers' comments:

Reviewer's Responses to Questions

**Comments to the Author**

1. If the authors have adequately addressed your comments raised in a previous round of review and you feel that this manuscript is now acceptable for publication, you may indicate that here to bypass the “Comments to the Author” section, enter your conflict of interest statement in the “Confidential to Editor” section, and submit your "Accept" recommendation.

Reviewer #1: All comments have been addressed

Reviewer #2: All comments have been addressed

2. Is the manuscript technically sound, and do the data support the conclusions?

Reviewer #1: Yes

Reviewer #2: Yes

3. Has the statistical analysis been performed appropriately and rigorously? 

Reviewer #1: N/A

Reviewer #2: N/A

4. Have the authors made all data underlying the findings in their manuscript fully available?

Reviewer #1: Yes

Reviewer #2: Yes

5. Is the manuscript presented in an intelligible fashion and written in standard English?

Reviewer #1: Yes

Reviewer #2: Yes

6. Review Comments to the Author

Reviewer #1: (No Response)

Reviewer #2: The authors did a good job of addressing the reviewer's comments.

Table 3.1 is long and difficult to follow. I think an updated table on the similarities and dissimilarities of the proposed guidelines will be easy to follow.

The second and third rows of Table 3.6 are confusing.

Do you need Section 3.5.1 in the manuscript? Even if you do, it should appear early in the manuscript. Preferably in a short form in the Introduction section.

Line 474 - 479: Please rethink. What chemicals in the vials contaminate the environment? I assume the major driver for management is the sharps?

7. PLOS authors have the option to publish the peer review history of their article (what does this mean?). If published, this will include your full peer review and any attached files.

Reviewer #1: No

Reviewer #2: No

---

## [Editor Report · Acceptance letter]

10 Aug 2022

PONE-D-21-41006R1 

Assessment of COVID-19 Vaccination-Related Medical Waste Management Practices in Bangladesh 

Dear Dr. Rahman:

I'm pleased to inform you that your manuscript has been deemed suitable for publication in PLOS ONE. Congratulations! Your manuscript is now with our production department. 

Kind regards, 

on behalf of

Dr. Nirupam Aich 

Academic Editor

PLOS ONE